# ZEPHYR GAN: REDEFINING GAN WITH FLEXIBLE GRADIENT CONTROL

## ABSTRACT

Generative adversarial networks (GANs) are renowned for their ability to generate highly realistic and diverse data samples. However, the performance of GANs is heavily dependent on the choice of loss functions, and commonly used losses such as cross-entropy and least squares are often susceptible to outliers, vanishing gradients, and training instability. To overcome these limitations, we introduce zephyr loss—a novel, convex, smooth, and Lipschitz continuous loss function designed to enhance robustness and provide flexible gradient control. Leveraging this new loss function, we propose ZGAN, a refined GAN model that guarantees a unique optimal discriminator and stabilizes the overall training dynamics. Furthermore, we demonstrate that optimizing ZGAN's generator objective minimizes a weighted difference between the real and generated data distributions. Through rigorous theoretical analysis, including convergence proofs, we substantiate the robustness and effectiveness of ZGAN, positioning it as a compelling and reliable alternative for stable GAN training. Extensive experiments further demonstrate that ZGAN surpasses leading methods in generative modeling.

## 1 INTRODUCTION

Generative adversarial networks (GANs) (Goodfellow et al., 2014) are a prominent generative modeling technique, where two neural networks engage in a competitive process: the discriminator, tasked with distinguishing real data from the generator's synthetic data, and the generator, which learns to produce realistic data from a noise source in order to deceive the discriminator. The theoretical aim of this competition is to achieve a Nash equilibrium. However, GANs may not consistently exhibit such equilibria. To address this, a proximal operator is applied to the original objective function, leading to the concept of proximal equilibrium (Farnia & Ozdaglar, 2020).

GANs have a wide range of real-world applications, including ClimateGAN (Schmidt et al., 2022), semantic image synthesis (Tang et al., 2023), text generation (Carlsson et al., 2024), and unsupervised pixel-wise anomaly detection (Mou et al., 2023). Since their inception, GANs (Bai et al., 2023; Sauer et al., 2021) have shown an exceptional ability to generate highly realistic and high-quality samples, making them valuable tools for various generative tasks. Numerous techniques have been introduced to enhance the training of GANs, such as Karras et al. (2019); Allen-Zhu & Li (2023); Asokan & Seelamantula (2023a); Saxena et al. (2023); Liu et al. (2023); Wang et al. (2024); Zhang et al. (2022).

Multiple versions are developed to enhance the stability and output quality of GAN. The slicing adversarial network (SAN) (Takida et al., 2024), for example, enhances gradient optimization through sliced optimal transport. Diffusion-GAN (Wang et al., 2023) improves the discriminator's effectiveness by generating Gaussian mixture noise via a forward diffusion mechanism. Additionally, the dynamically masked discriminator (Zhang et al., 2024) adapts to changes in generated data, while PCF-GAN (Lou et al., 2024) effectively captures temporal dependencies in time-series data. Spider GAN (Asokan & Seelamantula, 2023b), by leveraging structured images, enables faster convergence, thereby enhancing GAN performance across diverse applications. Further, GigaGAN (Kang et al., 2023) acts as a viable option for text-to-image synthesis.

The quality of GAN outputs is closely tied to the input distribution. Research indicates that low-dimensional latent vectors can effectively disentangle representations and control the characteristics

of the target being learned (Tran et al., 2017; Agustsson et al., 2019; Leśniak et al., 2019)). Further, deep generative networks (DGNs), utilized in both GANs and variational autoencoders (VAEs) (Kingma, 2013) to approximate data manifolds, can inadvertently reproduce biases present in the training data—stemming from preferences or convenience. This phenomenon can introduce artifacts that negatively impact fairness and other applications. To mitigate these biases, MaGNET (Humayun et al., 2022), a differential geometry-based sampler, has been proposed to generate uniformly distributed samples from any trained DGN, thus enhancing the quality and fairness of the generated outputs. Nowozin et al. (2016); Kurri et al. (2022) demonstrated that the generative-adversarial framework is a specific instance of a broader variational divergence approach, where any f-divergence can guide the training of generative neural samplers.

As the field of GANs evolves, various loss functions have been explored to improve training effectiveness. Techniques such as Wasserstein GAN (WGAN) (Arjovsky et al., 2017) and least squares GAN (LSGAN) (Mao et al., 2017) address specific challenges in the training process. WGAN which employs weight clipping leads to optimization issues; however, this approach has been refined in WGAN-GP (Gulrajani et al., 2017; Adler & Lunz, 2018), which overcomes optimization limitation while introducing an additional parameter that requires careful tuning. This added complexity highlights the ongoing need for innovative methods to optimize GAN training and ensure the generation of high-quality, realistic samples. LSGAN, for instance, encounters the vanishing gradient problem when generated samples closely resemble real ones, as its quadratic loss function can lead to diminishing gradients that slow further learning. The key contributions of our work are as listed:

- We introduce a novel loss function for GANs, termed zephyr loss, characterized by its smoothness around zero, which ensures stable gradient flow during training and mitigates issues commonly associated with vanishing or exploding gradients.
- We demonstrate that minimizing the zephyr loss effectively reduces the difference between the real and generated data distributions, leading to improved convergence and enhanced model performance.
- We conducted extensive experiments on well-known datasets, including CIFAR-10, CIFAR-100, STL-10, and SVHN, utilizing various models such as LSGAN, WGAN, and Diffusion-GAN. Our evaluation emphasized the performance of the proposed ZGAN and Diffusion-ZGAN models, showcasing their ability to generate high-quality and diverse images.

## 2 RELATED WORK

The training process of GANs (Goodfellow et al., 2014) is framed as a competitive game between two networks: the generator and the discriminator. The generator generates samples from a noise input, aiming to create realistic data, while the discriminator's role is to distinguish between genuine data from the real distribution and the synthetic samples generated by the generator. This process utilizes sigmoid cross-entropy loss to update the networks, where the discriminator's goal is to maximize the difference between real and fake data, while the generator seeks to minimize it by "fooling" the discriminator. Assume $\mathbf{z}$ is drawn from a noise distribution like uniform or Gaussian having $p_z$ distribution. Denote $\mathbf{x}$ as a real data sample, with $p_r$ indicating the distribution of real data, and $p_g$ representing the distribution of the generated data. This mechanism is captured by the following min-max objective function between generator $G$ and discriminator $D$:

$$\min_G \max_D \ \mathbb{E}_{\mathbf{x}\sim p_r}[\log(D(\mathbf{x}))] + \mathbb{E}_{\mathbf{z}\sim p_\mathbf{z}}[\log(1 - D(G(\mathbf{z})))].$$

If the discriminator reaches optimal training, the difference between the generator's distribution $p_g$ and the real data distribution $p_r$ is quantified using the Jensen-Shannon divergence (JSD).

While GANs have achieved significant progress in image generation, their output quality is still lacking for certain realistic applications. Traditional GANs using sigmoid cross-entropy loss often suffer from vanishing gradients during training, hindering effective learning. To overcome this, alternative loss functions like least squares loss and Wasserstein loss have been introduced, which reduce the vanishing gradient issue. However, despite these improvements, least squares loss is sensitive to outliers and Wasserstein loss has optimization issues, preventing them from fully addressing the challenges in GAN training.

## 2.1 HUBER LOSS

The mathematical formulation of Huber loss function (Wang & Shao, 2023) is given by

$$L_H(a) = \begin{cases} a & \text{if } |a| \leq \gamma, \\ \gamma(2|a| - \gamma) & \text{if } |a| > \gamma, \end{cases}$$

where $\gamma$ is a small positive parameter. The Huber loss function is convex and smooth.

## 3 PROPOSED WORK

To overcome the limitations of existing loss functions used in GAN discriminators—such as cross-entropy, least squares, and Wasserstein loss, which are prone to vanishing gradients, training instability, and sensitivity to noise and outliers—this section introduces a novel loss function called zephyr loss. Zephyr loss aims to provide a more effective solution that mitigates the challenges of conventional loss, improving training dynamics, and enhancing the stability and performance of GANs. This advancement represents a significant step forward in generative modeling, with the potential to improve outcomes across various generative tasks.

### 3.1 PROPOSED ZEPHYR LOSS

The mathematical formulation of zephyr loss is given as follows:

$$\mathcal{L}(a) = \alpha(\sqrt{a^2 + \epsilon} - \sqrt{\epsilon}), \tag{1}$$

where $\epsilon$ is a small positive quantity and $\alpha$ is a scaling factor. We demonstrate the zephyr loss through Figure 1.

**Significance of $\epsilon$:** The parameter $\epsilon$ is crucial for regularization and robustness. It smooths the gradient near $a = 0$, preventing abrupt changes that could destabilize training, especially in GANs where sharp gradients often lead to divergence. By moderating the gradient near zero, $\epsilon$ helps avoid large updates, ensuring smoother convergence.

**Significance of $\alpha$:** The scaling factor $\alpha$ modulates the overall magnitude of the zephyr loss and its gradients. A larger $\alpha$ results in amplified gradient updates, which can accelerate learning while also increasing the risk of instability. This tunable flexibility is crucial for adapting the loss function to diverse datasets and tasks, ultimately enhancing model performance and ensuring robust training stability.

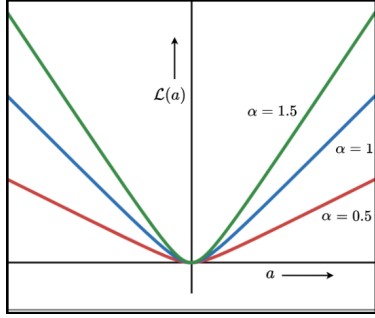

Figure 1: Representation of zephyr loss for different values of $\alpha$ for $\epsilon = 0.01$.

### 3.2 ZEPHYR GAN: METHODOLOGY

The utilization of novel zephyr loss in GAN training leads to the introduction of zephyr GAN abbreviated as ZGAN. The training and formulation of ZGAN for the case of discriminator and generator is given as shown:

$$\min_D O(D) = \mathbb{E}_{x \sim p_r}[\mathcal{L}(D(\mathbf{x}) - t)] + \mathbb{E}_{z \sim p_z}[\mathcal{L}(D(G(\mathbf{z})) - f)]$$

$$= \mathbb{E}_{x \sim p_r}[\alpha(\sqrt{(D(\mathbf{x}) - t)^2 + \epsilon} - \sqrt{\epsilon})] + \mathbb{E}_{z \sim p_z}[\alpha(\sqrt{(D(G(\mathbf{z})) - f)^2 + \epsilon} - \sqrt{\epsilon})],$$

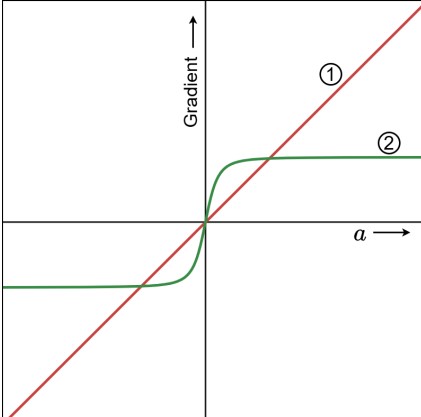

Figure 2: The figure illustrates the gradient behaviour of the zephyr loss, compared to the gradient of the least squares loss. Curve 1 corresponds to the gradient of the least squares loss. Curve 2 depict the gradient of zephyr loss, $\frac{\alpha a}{\sqrt{a^2+\epsilon}}$, with $\alpha = 0.5$, and $\epsilon = 0.01$.

where $t$ and $f$ represent the labels of true and fake data, which are 1 and 0, respectively. This objective encourages the discriminator to minimize the zephyr loss which is used to measure the closeness of the discriminator's output to the target values for both real and generated data. It effectively guides the discriminator to make accurate distinctions while maintaining stability in the gradient flow.

$$\min_G O(G) = \mathcal{L}(\mathbb{E}_{x \sim p_r}[D(\mathbf{x})] - \mathbb{E}_{z \sim p_z}[D(G(\mathbf{z}))])$$

$$= \alpha \left( \sqrt{\left( \int_{\mathbf{x}} p_r(\mathbf{x})D(\mathbf{x})dx - \int_{\mathbf{z}} p_z(\mathbf{z})D(G(\mathbf{z}))dz \right)^2 + \epsilon} - \sqrt{\epsilon} \right)$$

$$= \alpha \left( \sqrt{\left( \int_{\mathbf{x}} p_r(\mathbf{x})D(\mathbf{x})dx - \int_{\mathbf{x}} p_g(\mathbf{x})D(\mathbf{x})dx \right)^2 + \epsilon} - \sqrt{\epsilon} \right). \tag{2}$$

In the above formulation, the generator is designed to closely align the generated samples with the real data distribution, ensuring outputs that are statistically similar to real data. This approach reduces the risk of generating unrealistic or low-quality samples while improving the model's ability to capture the full diversity of the real data distribution, mitigating the risk of mode collapse. In other words, the generator's training strategy in ZGAN (equation 2) leverages the powerful technique of feature matching (Salimans et al., 2016a) to elevate its performance. By focusing on aligning with the internal representations of real data, rather than just tricking the discriminator, feature matching effectively combats mode collapse, preventing the generator from producing redundant outputs.

### 3.3 BENEFITS OF ZEPHYR LOSS

- **Enhanced gradient stability and balanced performance:** Zephyr loss enhances gradient stability by using a square root term to regulate the flow of gradients, particularly when the differences between real and generated data are small. This prevents vanishing or exploding gradients, common in losses like cross-entropy and least squares, resulting in smoother updates and more stable optimization, as demonstrated in Figure 2. Further, by moderating gradient updates for both generator and discriminator, zephyr loss balances their performance, avoiding instability, and ensuring that the generator produces diverse, outputs without allowing the discriminator to become overly dominant.

- **Enhanced robustness to outliers:** Zephyr loss, due to its formulation with the $\epsilon$ term, reduces the model's sensitivity to outliers or noisy data. For values of $a$ in equation 1 outside the range $[-\sqrt{\epsilon}, \sqrt{\epsilon}]$, the loss behaves more like the absolute difference, while for small values, it acts as a smoother function. This ensures that the training process is less

affected by extreme deviations in the data, allowing the model to generalize better and maintain consistent performance even when the training data contains noise.

- **Flexibility in tuning with $\alpha$:** The scaling factor $\alpha$ in zephyr loss allows for precise tuning of the loss magnitude. This flexibility is critical when working with different datasets and tasks, as the model can adapt to varying levels of difficulty or noise in the data. By adjusting $\alpha$, practitioners can control how aggressively the model learns, ensuring that the loss function contributes optimally to the GAN's overall performance.

Therefore, zephyr loss offers key advantages like gradient stability, robustness to noise, balanced training dynamics, and tunability, all of which contribute to more reliable and effective GAN training.

### 3.4 ZEPHYR LOSS VS HUBER LOSS

The Huber loss (Patterson et al., 2022) transitions between quadratic and linear regions based on a predefined threshold, while the Zephyr loss achieves this transition smoothly through its square root term, controlled by the parameter $\epsilon$, eliminating the need for an explicit threshold. Unlike the piecewise-defined Huber loss, the Zephyr loss is formulated as a single, continuous, and differentiable function, ensuring smoother gradient flow during optimization—an essential advantage in applications like adversarial learning where stability is critical. For large residuals ($|x|$), both losses exhibit linear growth; however, the Huber loss has a fixed slope determined by the threshold parameter, whereas the Zephyr loss allows direct control of the slope through the parameter $a$, providing greater flexibility.

## 4 THEORETICAL PROPERTIES

To gain deeper insights into the effectiveness and behaviour of proposed ZGAN model, we provide a thorough theoretical analysis. By examining divergences between probability distributions, we aim to offer a comprehensive understanding of the ZGAN framework. We also present a convergence theorem for ZGAN, further elucidating its performance characteristics.

**Theorem 1.** *The zephyr loss $\mathcal{L} : \mathbb{R} \to \mathbb{R}$ is smooth, convex and Lipschitz continous $\forall\, x \in \mathbb{R}$.*

Its continuity ensures that small changes in the model parameters lead to gradual and predictable variations in the loss, fostering stable training dynamics. The differentiability of zephyr loss allows for the effective calculation of gradients, facilitating efficient optimization through gradient-based methods. Further, being Lipschitz continuous, it provides a bounded rate of change, preventing issues like exploding or vanishing gradients, which enhances robustness during training. The convexity of the loss ensures that gradient-based optimization moves steadily toward a global minimum, promoting consistent progress in training. Collectively, these properties create a smooth loss landscape, which fosters reliable convergence and improves the overall performance of generative models, especially in the challenging context of adversarial training.

**Theorem 2.** *If $a \in (-\sqrt{\epsilon}, \sqrt{\epsilon})$, then the proposed zephyr loss defined in equation (1) approximately reduces to the least squares loss.*

*Proof.* $\mathcal{L}(a) = \alpha(\sqrt{a^2 + \epsilon} - \sqrt{\epsilon}) = \alpha(\sqrt{\epsilon}\sqrt{(1 + \frac{a^2}{\epsilon})} - \sqrt{\epsilon}) \underset{\frac{a^2}{\epsilon} < 1}{\approx} \alpha(\sqrt{\epsilon} + \frac{a^2}{2\sqrt{\epsilon}} - \sqrt{\epsilon}) = \alpha\frac{a^2}{2\sqrt{\epsilon}}.$

Thus, by binomial expansion which holds when $\frac{a^2}{\epsilon} < 1$, that is, $a \in (-\sqrt{\epsilon}, \sqrt{\epsilon})$, the zephyr loss can be approximated by least squares loss. $\qquad\square$

**Theorem 3.** *For a fixed generator $G$, the optimal discriminator $D^*$ is unique.*

*Proof.* The optimization problem of the discriminator is given by:

$$\min_D O(D) = \int_x p_r(\mathbf{x})\alpha(\sqrt{(D(\mathbf{x}) - t)^2 + \epsilon} - \sqrt{\epsilon})dx + \int_z p_z(\mathbf{z})\alpha(\sqrt{(D(G(\mathbf{z})) - f)^2 + \epsilon} - \sqrt{\epsilon})dz$$

$$= \int_x (p_r(\mathbf{x})\alpha(\sqrt{(D(\mathbf{x}) - t)^2 + \epsilon} - \sqrt{\epsilon}) + p_g(\mathbf{x})\alpha(\sqrt{(D(\mathbf{x}) - f)^2 + \epsilon} - \sqrt{\epsilon}))dx.$$

$$(3)$$

The goal is to minimize $O(D)$ with respect to $D$. The function is strictly convex with respect to $D$ for $p_r, p_g \in (0, \infty)$. According to the properties of strictly convex functions, any strictly convex function possesses a unique global minimum. Therefore, the optimal discriminator $D^*$ minimizing $O(D)$ is unique. $\qquad \square$

**Corollary 3.1.** *If $p_r(\mathbf{x}) = p_g(\mathbf{x})$, then the optimal discriminator is given by $D^*(\mathbf{x}) = \frac{1}{2}$, for a fixed generator.*

*Proof.* Consider the discriminator's objective function, $O(D)$, as defined in Equation (3). To find the optimal discriminator $D^*(\mathbf{x})$, we compute the derivative of $O(D)$ with respect to $D$, set it to zero, and solve for $D^*(\mathbf{x})$. Under the condition $p_r(\mathbf{x}) = p_g(\mathbf{x})$, solving the resulting equation yields $D^*(\mathbf{x}) = \frac{1}{2}$. $\qquad \square$

**Theorem 4.** *Let $D^*$ be the optimal discriminator for a fixed generator $G$. Minimizing the generator's objective function is a regularized approximation of the Wasserstein-1 ($W1$) distance.*

*Proof.* Consider the generator's objective function $C(G)$ as defined in equation (2):

$$\min_G C(G) = \alpha \left( \sqrt{\left( \int_{\mathbf{x}} p_r(\mathbf{x}) D^*(\mathbf{x}) dx - \int_{\mathbf{z}} p_z(\mathbf{z}) D^*(G(\mathbf{z})) dz \right)^2 + \epsilon} - \sqrt{\epsilon} \right)$$

$$= \alpha \left( \sqrt{\left( \int_{\mathbf{x}} p_r(\mathbf{x}) D^*(\mathbf{x}) dx - \int_{\mathbf{x}} p_g(\mathbf{x}) D^*(\mathbf{x}) dx \right)^2 + \epsilon} - \sqrt{\epsilon} \right)$$

$$= \alpha \left( \sqrt{\left( \int_{\mathbf{x}} p_r(\mathbf{x}) D^*(\mathbf{x}) dx - \int_{\mathbf{x}} p_g(\mathbf{x}) D^*(\mathbf{x}) dx \right)^2 + \epsilon} \right). \qquad (4)$$

For small values of $\epsilon$, we can approximate the function $C(G)$ as follows:

$$C(G) \approx \alpha \left| \int_{\mathbf{x}} (p_r(\mathbf{x}) - p_g(\mathbf{x})) D^*(\mathbf{x}) dx \right|.$$

Minimizing the generator's objective function is equivalent to minimizing the difference between the real and generated data distributions, where the difference is weighted by the optimal discriminator function $D^*(\mathbf{x})$. This difference is represented as the integral of the difference between the real data distribution $p_r(\mathbf{x})$ and the generated data distribution $p_g(\mathbf{x})$, with the weighting factor $D^*(\mathbf{x})$ over the input space $x$.

$\qquad \square$

**Theorem 5.** *The objective function $C(G)$ attains its global minimum value of 0 if and only if the real data distribution $p_r$ is equal to the generated data distribution $p_g$.*

*Proof.* If $p_r(\mathbf{x}) = p_g(\mathbf{x})$, it follows from equation (4) that $C(G) = 0$, representing the minimum value attainable by the objective function $C(G)$. Conversely, if $C(G) = 0$, then $p_r(\mathbf{x}) = p_g(\mathbf{x})$.
**Note:** $O'(D) = \frac{p_r(\mathbf{x})(D(\mathbf{x})-t)}{\sqrt{(D(\mathbf{x})-t)^2+\epsilon}} + \frac{p_g(\mathbf{x})(D(\mathbf{x})-f)}{\sqrt{(D(\mathbf{x})-f)^2+\epsilon}} \implies O'(0) = \frac{p_r(\mathbf{x})(-t)}{\sqrt{t^2+\epsilon}} + \frac{p_g(\mathbf{x})(-f)}{\sqrt{(f)^2+\epsilon}} \neq 0$, for $p_g, p_r \in (0, 1]$, $t = 1, f = 0$. Therefore, $D^*(\mathbf{x}) \neq 0$, that is, the optimal discriminator cannot be zero. $\qquad \square$

**Proposition 1.** *Let the discriminator $D$ and generator $G$ be sufficiently expressive. If the discriminator is allowed to reach its optimal state $D^*$ for a given generator $G$, and the generator's distribution $p_g$ is updated by optimizing the criterion*

$$\alpha \left( \sqrt{(p_r(\mathbf{x}) D^*(\mathbf{x}) - p_g(\mathbf{x}) D^*(\mathbf{x}))^2 + \epsilon} - \sqrt{\epsilon} \right),$$

*then the generated distribution $p_g$ converges to the real data distribution $p_r$ as the training progresses.*

*Proof.* Consider the objective function $O(G)$ from equation (2) as a function of the generator's distribution $p_g$. The second derivative of $O(G)$ with respect to $p_g$ is given by $\frac{\epsilon(D(\mathbf{x}))^2}{\left((p_r(\mathbf{x})D(\mathbf{x})-p_g(\mathbf{x})D(\mathbf{x}))^2+\epsilon\right)^{\frac{3}{2}}}$, which is non-negative. Therefore, the objective function $O(G)$ is convex with respect to $p_g$. The convexity ensures that the optimization landscape of $O(G)$ has global minima. The subderivative of $O(G)$ at a given $G$ provides the direction of steepest descent, which governs the updates to $p_g$. When the discriminator $D$ is at its optimal state $D^*$, this subderivative corresponds to the gradient descent update for $p_g$. Since the infimum of a family of convex functions is itself convex, the function $O_{D^*}(G)$ remains convex with respect to $p_g$. As the training process advances, the generator $G$ continues to optimize $O_{D^*}(G)$, which will converge towards the global minimum. Consequently, from Theorem 5 the convergence of $G$ towards optimality leads to the generator's distribution $p_g$ approximating the real data distribution $p_r$. □

## 5 EXPERIMENTS

In this section, we present the experimental setup used to evaluate the performance of proposed ZGAN. First, we provide an overview of the datasets employed in our study, detailing the specific characteristics of each dataset. We then describe the implementation details, including the network architecture, hyperparameters, and training protocols. Finally, we assess the effectiveness of ZGAN through experiments conducted on multiple scene datasets, comparing the results with existing models and highlighting key performance metric.

**Datasets:** We conducted experiments using several image datasets, including CIFAR-10 ($32 \times 32$) (Krizhevsky & Hinton, 2009), CIFAR-100 ($32 \times 32$) (Krizhevsky & Hinton, 2009), STL-10 ($96 \times 96$) (Coates et al., 2011), and SVHN ($32 \times 32$) (Netzer et al., 2011). These datasets are chosen for their varying image sizes, ranging from $28 \times 28$ to $96 \times 96$ pixels, allowing us to evaluate the performance of proposed models across a diverse set of data characteristics. By including datasets with different resolutions and complexities, we aimed to thoroughly assess the robustness and generalization capabilities of the models in generating high-quality images.

### 5.1 EXPERIMENTAL FRAMEWORK

We conducted a thorough comparison of the proposed ZGAN model with LSGAN and WGAN. To extend our evaluation, we also compared ZGAN with Diffusion-GAN (Wang et al., 2023), which incorporates a diffusion process for improved stability and sample quality. Expanding on this approach, we introduce a novel variant, Diffusion-ZGAN, which adapts the diffusion process from the Diffusion-GAN framework to the ZGAN setting. In Diffusion-ZGAN, we apply a diffusion process to the input images before feeding them into the discriminator. This is achieved by adding noise to both real and generated images, ensuring that the discriminator evaluates the images after they have undergone the diffusion process. All the models follow the publicly available implementation of deep convolutional GANs (DCGANs)[1] (Radford, 2015), utilizing the PyTorch framework. This allowed us to leverage well-established architectures while incorporating the proposed zephyr loss function and model.

We performed an extensive experimental study across multiple datasets, evaluating the performance of different models while maintaining a consistent architecture for both the generator and discriminator in each case. To ensure a fair and unbiased comparison, we applied the same set of hyperparameters for all models, unless otherwise specified. Specifically, for all models except WGAN, we used Adam optimizer (Kingma, 2014) with a learning rate of $0.0002$ and a $\beta_1$ value of $0.5$. For WGAN, as recommended by (Arjovsky et al., 2017), we employed a RMSProp (Tieleman, 2012), using a clip value of $0.01$ and a learning rate of $0.0005$. These adjustments are made to promote stable and efficient training, addressing the unique requirements of the WGAN framework.

In the proposed ZGAN and Diffusion-ZGAN, we conducted extensive tuning of the parameters $\alpha$ and $\epsilon$ to optimize its performance. We explored a range of $\alpha$ values, specifically 0.3, 0.5, 0.8, 1, and 1.5, and found through empirical testing that the model achieved the best results when $\alpha$ was set to 0.5 for the discriminator and 1 for the generator. Additionally, we investigated the impact of $\epsilon$ in the loss function, testing values of 0.1, 0.01, 0.001, and 0.0001. Our experiments revealed that

---

[1]https://github.com/Lornatang/DCGAN-PyTorch

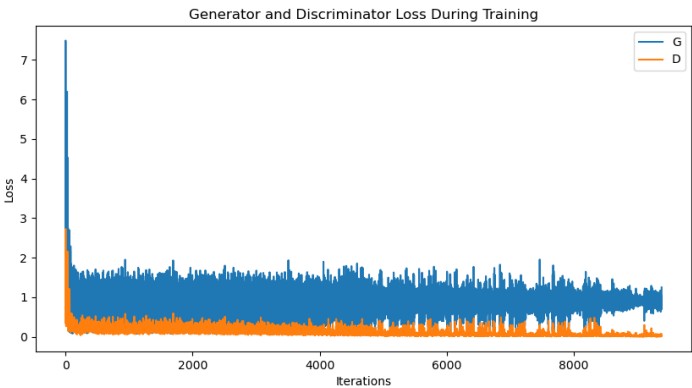

Figure 3: Training dynamics of ZGAN: Evolution of discriminator and generator losses across iterations.

ZGAN performed optimally with an $\epsilon$ value of 0.01. Based on these findings, we adopted an $\alpha$ value of 0.5 for the discriminator, 1 for the generator, and an $\epsilon$ value of 0.01 in the final ZGAN model. Throughout all experiments, we consistently maintained a batch size of 64 for training across all models to ensure uniformity and reliable comparisons. We assigned values of 1 and 0 for the true and fake labels, $t$ and $f$, respectively (shown in equation 3).

For the quantitative assessment of models, we employed the inception score (Salimans et al., 2016b) as a primary metric, which is a recognized metric for assessing the quality and diversity of generated images. A higher inception score indicates that the generated images possess both high quality and diversity, reflecting a broad range of categories which makes the inception score an essential metric for evaluation. To ensure the reliability of our experiments, we conducted all tests on a robust computing infrastructure equipped with NVIDIA A4500 GPUs.

Table 1: Inception score of proposed ZGAN and Diffusion-ZGAN, along with baseline models Diffusion-GAN, LSGAN and WGAN over benchmark datasets: CIFAR-10, CIFAR-100, STL-10, and SVHN. The best values are represented in bold.

| Datasets | WGAN | LSGAN | Diffusion-GAN | ZGAN* | Diffusion-ZGAN* |
|----------|------|-------|---------------|-------|-----------------|
| CIFAR-10 | 1.11 | 2.1 | 2.63 | 2.85 | **2.9** |
| CIFAR-100 | 1.1 | 2.69 | 2 | 2.54 | **3.01** |
| STL-10 | 1.2 | 1.54 | 2.13 | 1.76 | **2.23** |
| SVHN | 1.12 | 2 | **2.19** | 1.5 | 2.13 |

* denote proposed models.

### 5.2 QUANTITATIVE AND QUALITATIVE ANALYSIS OF RESULTS

The quantitative results of a comprehensive series of experiments in Table 1, demonstrate the superiority of both ZGAN and Diffusion-ZGAN over baseline models. These models, leveraging the zephyr loss, show enhanced image generation capabilities, producing more diverse and realistic outputs across various datasets. The results demonstrate that zephyr loss enhances the quality of generated images, providing a robust and reliable alternative to traditional loss functions. Figure 3 illustrates the dynamic balance between the discriminator ($D$) and generator ($G$) during ZGAN training. The figure shows that the losses of both $D$ and $G$ oscillate, indicating mutual improvement. If there is a significant drop in $D$'s loss while $G$'s loss rises may signal an imbalance, where the discriminator becomes too strong, hindering the generator's learning. Referring to Figure 3, as the losses for both $D$ and $G$ stabilize and oscillate around specific values, it indicates that the GAN is reaching an equilibrium. This balanced state between $D$ and $G$ suggests that the model is effectively generating realistic samples. The combination of enhanced performance and stability underscores the ability of the proposed models to address the prevalent challenges in GAN training effectively.

Figure 4 illustrates a qualitative comparison between images generated by LSGAN and the proposed ZGAN across the CIFAR-10, CIFAR-100, STL-10, and SVHN datasets. ZGAN consistently produces sharper, more detailed images with improved diversity, capturing fine-grained patterns and complex textures inherent in each dataset. In contrast, the images generated by LSGAN exhibit relatively less clarity and variation, often lacking finer details. ZGAN's outputs display a stronger grasp of structural and visual elements, particularly in more challenging datasets like CIFAR-100 and STL-10, indicating a superior ability to model diverse and intricate data representations.

In order to thoroughly evaluate the performance of the proposed models, we utilize well-established metrics like the Fréchet Inception Distance (FID) (Lucic et al., 2018) and Kernel Inception Distance (KID) (Sutherland et al., 2018), which provide valuable insights into the quality and diversity of the generated images. The results presented in Tables 2 and 3 summarize the FID and KID scores of various generative models, including our proposed models, ZGAN and Diffusion-ZGAN, alongside baseline models such as WGAN, LSGAN, and Diffusion-GAN. The comparison highlights that both ZGAN and Diffusion-ZGAN consistently outperform the baseline models in terms of both FID and KID, indicating their superior ability to generate high-quality and diverse images. Notably, Diffusion-ZGAN achieves particularly strong performance on benchmark datasets such as CIFAR10 and SVHN, demonstrating a significant reduction in FID and KID scores compared to the baseline models. This indicates that Diffusion-ZGAN excels at capturing the underlying distribution of the real data, producing generated images that are both realistic and diverse.

On the other hand, ZGAN also demonstrates competitive results, especially on the CIFAR100 dataset, where it performs comparably to other state-of-the-art models. This reinforces the versatility of ZGAN across different datasets, showing that it can be a strong contender for generating diverse images in various domains. These results underscore the effectiveness of the proposed models in generating images that not only closely resemble real images but also exhibit a high degree of diversity.

| FID Score | WGAN | LSGAN | Diffusion-GAN | ZGAN | Diffusion-ZGAN |
|---|---|---|---|---|---|
| CIFAR10 | 1.912 | **1.54** | 2.02 | 3.003 | **1.54** |
| CIFAR100 | 1.897 | **1.5** | 2.6 | 2.65 | 1.66 |
| STL10 | 2.634 | 2.6 | **2.52** | 3.48 | 3.54 |
| SVHN | 15.94 | 17.49 | 64.53 | **10.39** | 16.94 |

Table 2: FID score of proposed ZGAN and Diffusion-ZGAN, along with baseline models Diffusion-GAN, LSGAN and WGAN over benchmark datasets: CIFAR-10, CIFAR-100, STL-10, and SVHN. The best values are represented in bold.

| KID Score | WGAN | LSGAN | Diffusion-GAN | ZGAN | Diffusion-ZGAN |
|---|---|---|---|---|---|
| CIFAR10 | 0.229 | 0.172 | 0.163 | 0.372 | **0.161** |
| CIFAR100 | 0.17 | 0.089 | 0.11 | 0.321 | **0.11** |
| STL10 | 0.181 | 0.177 | **0.141** | 0.474 | 0.475 |
| SVHN | 0.18 | 0.283 | 0.176 | 0.169 | **0.175** |

Table 3: KID score of proposed ZGAN and Diffusion-ZGAN, along with baseline models Diffusion-GAN, LSGAN and WGAN over benchmark datasets: CIFAR-10, CIFAR-100, STL-10, and SVHN. The best values are represented in bold.

# 6  ABLATION STUDY ON HYPERPARAMETERS $\epsilon$ AND $\alpha$

We conducted a detailed ablation study on the CIFAR-10 dataset to assess the impact of the two key hyperparameters of the Zephyr loss function, $\epsilon$ and $\alpha$, on the performance of ZGAN. To illustrate the relationship between these hyperparameters and model performance, we present the FID scores for different values of $\alpha$ and $\epsilon$ in Tables 4 and 5, respectively. Our experiments reveal that both $\epsilon$ and $\alpha$ have a significant effect on the performance of the loss function. Specifically, as $\epsilon$ increases from 0.0001 to 0.01, we observe a marked improvement in the FID score. This suggests that larger values of $\epsilon$ improve gradient stability during training, which is essential for effective optimization.

A small $\epsilon$ value results in less gradient stabilization, making the training process more susceptible to small fluctuations in the gradients, which can adversely impact performance. Thus, we conclude that $\epsilon$ plays a critical role in ensuring smoother training dynamics and overall model robustness.

In terms of $\alpha$, the optimal performance, as measured by the FID score, is achieved at $\alpha = 0.5$, which strikes the best balance. The large values of $\alpha$ lead to instability, reducing the robustness of the model and causing a drop in performance.

In summary, the results of our ablation study indicate that both very small ($\alpha$ causing vanishing gradients) and very large ($\alpha$ leading to exploding gradients) values of $\alpha$ are suboptimal. Similarly, too small value of $\epsilon$ fails to provide sufficient stability during training.

| $\epsilon$ | 0.0001 | 0.001 | 0.01 |
|---|---|---|---|
| FID | 2.72 | 2.64 | **2.15** |

Table 4: Ablation Study on the parameter $\epsilon$ for the proposed ZGAN on CIFAR10 dataset.

| $\alpha$ | 0.5 | 0.8 | 1 |
|---|---|---|---|
| FID | **1.88** | 2.69 | 2.58 |

Table 5: Ablation study on parameter $\alpha$ for the proposed ZGAN on CIFAR10 dataset.

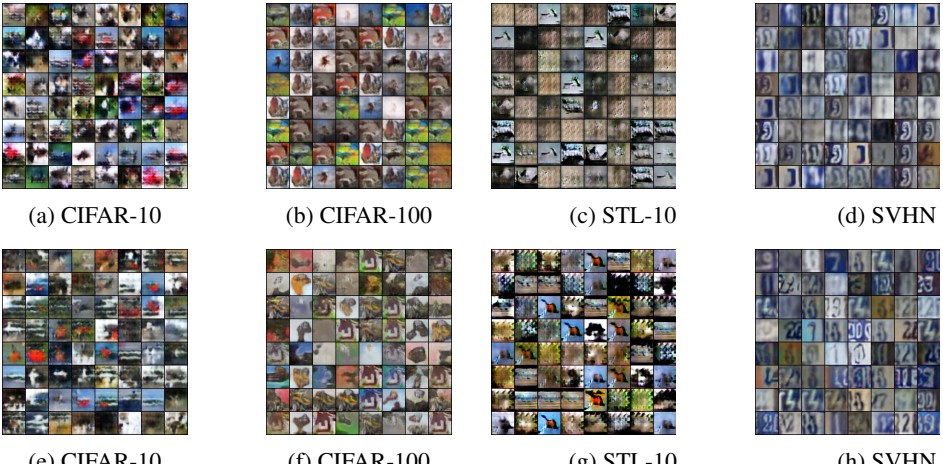

| (a) CIFAR-10 | (b) CIFAR-100 | (c) STL-10 | (d) SVHN |
|---|---|---|---|
| (e) CIFAR-10 | (f) CIFAR-100 | (g) STL-10 | (h) SVHN |

Figure 4: Images generated by the LSGAN and proposed ZGAN models. The first row displays images by LSGAN, while the second row presents images by proposed ZGAN.

## 7 CONCLUSION

This work unveils a novel loss function for generative adversarial networks (GANs) named zephyr loss. Characterized by its smoothness, convexity, and Lipschitz continuity, zephyr loss significantly transforms the landscape of GAN training. Leveraging the transformative capabilities of zephyr loss, we introduce the ZGAN model, which guarantees a unique optimal discriminator. We provide rigorous theoretical proof of the convergence of the proposed ZGAN model, solidifying its standing in the realm of GAN research. Our extensive experimental evaluations illustrate the superior performance of ZGAN in comparison to existing models that utilize different loss functions. The results demonstrate that ZGAN exemplifies reliability and stability in GAN training, marking the onset of a new era in generative modeling. Moreover, by seamlessly integrating the diffusion process into the ZGAN framework, we present Diffusion-ZGAN, further amplifying the robustness and efficacy of our approach. ZGAN advances the current state of GANs, laying a strong foundation for future research in generative modeling. However, the advantages of employing Zephyr loss may be compromised if the feature representations lack sufficient informativeness. While Zephyr loss demonstrates robust performance on standard benchmark datasets, its scalability to large-scale or high-dimensional datasets remains an area for future investigation. Moreover, there is significant potential for extending this work to vector quantized GANs (Esser et al., 2021; Yu et al., 2022), further enhancing its applicability across diverse generative tasks.

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

## A  APPENDIX

### A.1  THEORETICAL OBSERVATION

**Strong convexity (Zhou, 2018)** If a function $f$ is twice continuously differentiable and defined on the real line, it is strongly convex if and only if $f''(x) \geq m > 0$ for all $x$.

**Theorem 6.** $O(D)$ *is strongly convex with respect to* $D$.

*Proof.* To demonstrate that $O(D)$ is strongly convex, we evaluate the second derivative of $O(D)$ with respect to $D$. The second derivative is given by:

$$\alpha\epsilon\left(\frac{p_r(\mathbf{x})}{(D(\mathbf{x})-t)^2+\epsilon)^{3/2}} + \frac{p_g(\mathbf{x})}{(D(\mathbf{x})-f)^2+\epsilon)^{3/2}}\right).$$

The terms $(D(\mathbf{x})-t)$ and $(D(\mathbf{x})-f)$ are the differences between the discriminator's predicted label $D(\mathbf{x})$ and the true and generated labels $t$ and $f$. These differences are finite and bounded. Consequently, the denominators $(D(\mathbf{x})-t)^2+\epsilon)^{3/2}$ and $(D(\mathbf{x})-f)^2+\epsilon)^{3/2}$ are strictly positive for $\epsilon > 0$, ensuring they do not approach zero. Further, since $(D(\mathbf{x})-t)$ and $(D(\mathbf{x})-f)$ are finite, the denominators are also bounded above.

These bounds ensure that the second derivative is strictly positive for all valid $D$ as $\alpha\epsilon > 0$, $p_r(\mathbf{x})$ and $p_g(\mathbf{x}) > 0$. Therefore, $O(D)$ satisfies the condition for strong convexity with respect to $D$.

□

## B  ZEPHYR LOSS VERSUS CONVENTIONAL LOSS FUNCTIONS

In this section, we compare the proposed zephyr loss with the well-known existing loss functions in the context of GAN training.

### B.1  ZEPHYR LOSS VS CROSS-ENTROPY LOSS

In GAN, the cross-entropy loss is used in the discriminator for binary classification between real and fake samples, however, it leads to vanishing gradients when the discriminator becomes overly confident, which impedes the generator's learning (Radford, 2015). Furthermore, cross-entropy loss does not consider the distance between the discriminator's output and the true/fake labels, focusing solely on correct classifications. In contrast, zephyr loss addresses these shortcomings by incorporating a measure of this distance, allowing for more informative gradient updates. This approach not only maintains a smoother gradient flow but also ensures that the generator continues to receive meaningful feedback, facilitating continuous learning and enhancing training stability.

### B.2  ZEPHYR LOSS VS LEAST SQUARES LOSS

In LSGAN, the quadratic loss over-penalizes outliers, leading to inefficient learning and a focus on rare samples, while also causing vanishing gradients as the discriminator becomes more confident, slowing the generator's progress. In contrast, zephyr loss is designed to maintain smoother gradients and gradient control throughout training, preventing the vanishing gradient problem and ensuring more balanced updates. This allows ZGAN to achieve stable and efficient training without overfocusing on outliers, while also encouraging continuous improvement in the generator's outputs without stalling the learning process. In the least square loss $l(a) = a^2$, the gradient $l'(a) = 2a$ is directly proportional to $a$, where $a$ represents the difference between the discriminator's predicted label and the true or fake label. This linear dependence on $a$ can result in gradient instability during training. Specifically, when $a$ becomes very small, the gradient $l'(a)$ diminishes rapidly, leading to slower convergence or even stalling the training process. Conversely, for large values of $a$, which may occur in the presence of outliers, the gradient $l'(a)$ becomes excessively large, causing abrupt updates and making the model sensitive to noise and outliers. ZGAN, on the other hand, introduces

the zephyr loss $\mathcal{L}(a) = \alpha(\sqrt{a^2 + \epsilon} - \sqrt{\epsilon})$, where $\epsilon > 0$ is a small regularization parameter. The gradient of the zephyr loss is given by:

$$\mathcal{L}'(a) = \frac{a}{\sqrt{a^2 + \epsilon}}.$$

This gradient has the following key properties: It stabilizes for large $a$, as $\mathcal{L}'(a) \to 1$, preventing overly large updates caused by outliers. For small $a$, the rate at which $\mathcal{L}'(a) \to 0$ is slower than the gradient of the least square loss $l'(a)$, which diminishes linearly. In the case of zephyr loss, the presence of $\epsilon$ in the denominator ensures that the gradient diminishes more smoothly. These characteristics of the zephyr loss ensure smoother and more controlled updates to the model parameters, leading to greater stability and robustness during training.

### B.3 ZGAN vs WGAN

WGAN employs weight clipping to maintain the Lipschitz condition, which can limit the critic's (discriminator of WGAN) capacity and hinder the generator's learning. WGAN-GP addresses this with a gradient penalty, improving training stability but also increasing computational complexity and the need for careful hyperparameter tuning. In contrast, ZGAN controls the loss function shape to maintain smooth gradients and stable training. This approach allows ZGAN to enhance training efficiency and gradient flow.

## C FUTURE WORK

Higher-resolution datasets and modern GAN architectures, such as StyleGAN, are well-suited for generating high-quality images. While initial experiments to evaluate Zephyr loss within these advanced architectures have been initiated, time and hardware constraints have limited the ability to complete them fully. Nevertheless, preliminary results suggest that Zephyr loss integrates seamlessly into modern architectures and has the potential to further enhance the quality of generated images in these settings. To illustrate this, an image generated using Zephyr loss within the StyleGAN-XL framework on the CIFAR-10 dataset is provided in Figure 5, demonstrating the quality achievable with this approach. Future work will focus on extending these evaluations to high-resolution image generation with state-of-the-art architectures.

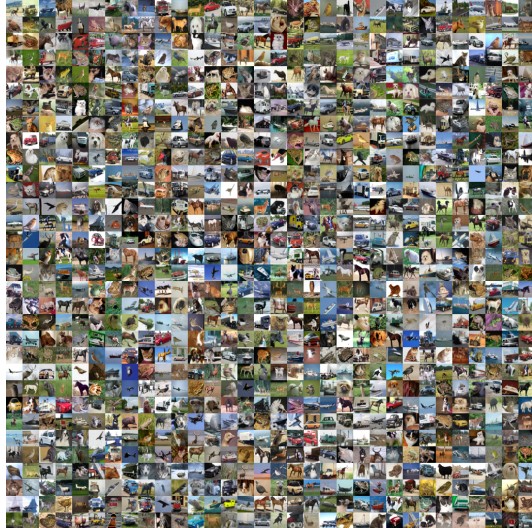

Figure 5: CIFAR10 dataset image generated by zephyr-StyleGAN-XL

