# OpenReview forum: "ZEPHYR GAN: REDEFINING GAN WITH FLEXIBLE GRADIENT CONTROL"
_ICLR.cc/2025/Conference — Submitted to ICLR 2025_

### Official Review · Reviewer_jaht · 2024-10-22

**Soundness:** 1
**Presentation:** 2
**Contribution:** 1
**Rating:** 1
**Confidence:** 5

**Summary:**

The author claims that the zephyr loss improve stability, prevent overfitting, and makes it robust to outliers. They provide some theorems about the properties of the loss and showing weak experimental results.

**Strengths:**

Novel loss function, could potential help stability (not enough proof for that claim yet). Needs a lot more work to validate it.

**Weaknesses:**

Theorem 2 is framed incorrectly, its approximately similar, not equal. It would be great to instead give lower and upper bounds.

As mentioned in WGAN, https://arxiv.org/abs/1701.07875, the Total Variation is not continuously differentiable with respect to the parameters. Since Zephyr loss is equivalent to estimating the Total Variation distance, this is not ideal.

The experiments alone prevent giving this paper a good score. The generated images are very poor quality, often completely broken and the inception score is extremely low (For CIFAR-10, the authors get an FID of 2.9 when the smallest FID of the WGAN-GP 7 years old paper is 5.34 and they can reach up to 8.59). We are not in 2016, currently they were done with the original setup from DCGAN. The experiments need to be done with modern implementations using Adamw(beta1=0), EMA, and modern methods. The baselines should be taken from their respective papers or re-implemented with modern methods with similar or better FID/IS than original results.

At this point in time the big problem with GANs is stability when scaling. Making a new loss function is unlikely to solve this problem. The authors needs better experiments and it would be useful to also have DiracGAN (https://github.com/ChristophReich1996/Dirac-GAN) experiments and convergence proof.

IS is not a good metric, this has been mentioned many times in the literature. FID is better, but not perfect. Still, its a better choice.

**Questions:**

Why do you use old techniques instead of modern ones?

---

### Official Review · Reviewer_qhtZ · 2024-10-30

**Soundness:** 2
**Presentation:** 2
**Contribution:** 2
**Rating:** 3
**Confidence:** 5

**Summary:**

This paper proposes a new loss function, zephyr loss, and applies it to the training of generative adversarial networks to stabilize the training and solve the problems of gradient vanishing and unstable training that existed in previous generative adversarial networks. Theoretical analysis and experiments are conducted to verify the effectiveness of this loss function.

**Strengths:**

1. The zephyr loss function is proposed to address common GAN issues (e.g., vanishing gradients, training instability). This is especially relevant given that the loss is smooth around zero, theoretically improving gradient stability and mitigating exploding/vanishing gradients.
2. The authors provide detailed proofs for the zephyr loss properties, including convexity, smoothness, and Lipschitz continuity, which contribute to ZGAN's stability.
3. The experiments on datasets (CIFAR-10, CIFAR-100, STL-10, SVHN) provide evidence that ZGAN outperforms traditional GANs (e.g., LSGAN, WGAN) in terms of inception scores, indicating higher quality and diversity in generated images.

**Weaknesses:**

1. My primary concern is the Inadequate experimental evaluation conducted in the paper.
  • The authors rely solely on the Inception Score (IS) to assess the performance of different GANs, which is insufficient for a comprehensive evaluation. Additional metrics, such as FID and KID, should be incorporated to provide a more reasonable comparison. The FID compares the distribution of generated images to that of real images in the feature space of the Inception model. It captures both the quality and diversity by measuring the distance between the two distributions. FID is particularly useful because it correlates well with human judgment of image quality, making it a more reliable indicator of GAN performance. KID is similar to FID but is based on a different statistical approach. It calculates the squared maximum mean discrepancy between the feature distributions of generated and real images. One of the advantages of KID is that it is unbiased and can be applied to small sample sizes, which makes it useful in scenarios where the number of generated images is limited. incorporating FID and KID alongside Inception Score enables a more nuanced evaluation of GAN performance, addressing the shortcomings of relying on a single metric and providing insights into both the realism and diversity of the generated outputs.
  • All experiments are conducted using relatively simple networks and datasets. To strengthen the validation of the proposed loss function, the authors are suggested to employ the proposed loss function to some more advanced architectures like StyleGAN2 and StyleGAN-XL, and test them on more challenging datasets such as FFHQ and AFHQ. StyleGAN2 and StyleGAN-XL are the most popular and advanced GANs. It is necessary to verify the proposed loss function based on these advanced models and challenging datasets.
  • Since the paper focuses on the stable training of GANs, it is well known that GANs are even more unstable when trained on a small amount of data. Authors are suggested to conduct experiments with a small dataset to verify whether the proposed loss function can also stabilize GAN training on small amount of training data. Obama-100, Sketch-100, and FFHQ-Sunglasses are three common few-shot image generation datasets. By focusing on these aspects, you can get a clearer picture of the model's stability and robustness.
  • The authors only compare the proposed method with WGAN and LSGAN. More comparison with WGAN-GP, WGAN-LP are expected.
  • Note that the proposed loss function contains two variables, the authors should conduct ablation experiments on these two variables and list the experimental results in the form of figures or lists, rather than just briefly mentioning them in the main text. Given the current experiment setup, it is difficult for me to be fully convinced of the performance advantages claimed for the proposed method.

2. This paper provides a proof of the Lipschitz continuity for the proposed loss function. However, as far as I am aware, other loss functions commonly used in GANs also share this property. Ensuring the continuity of the discriminator is fundamental to achieving stable GAN training.

3. The 5th section of the paper seems to be redundant, and is not supported by any experimental results. It might be better to place it in the analysis of experimental results in 6th section.

**Questions:**

1. Additional metrics, such as FID and KID, are suggested to be adopted to assess the performance of GANs in the paper.
2. The proposed loss function is suggested to employed to the StyleGAN2 and StyleGAN-XL, and compare them with more advanced models, e.g. StyleGAN2, StyleGAN-XL, WGAN-GP and WGAN-LP.
3. The ablation experiments on the two variables in the proposed loss function should be conducted, and the experimental results should be exhibited in figures or lists.
4. Authors are suggested to conduct training experiments with a small dataset to verify whether the proposed loss function can also stabilize GAN training on small amount of training data.

---

### Official Review · Reviewer_UUhK · 2024-11-01

**Soundness:** 3
**Presentation:** 2
**Contribution:** 2
**Rating:** 3
**Confidence:** 4

**Summary:**

To overcome the limitations such as susceptible to outliers, vanishing gradients, and training instability in GAN’s training, this paper introduce zephyr loss—a novel, convex, smooth, and Lipschitz continuous loss function designed to enhance robustness and provide flexible gradient control.
Leveraging this new loss function, the article propose ZGAN, a refined GAN model that guarantees a unique optimal discriminator and stabilizes the overall training dynamics.
Extensive experiments further demonstrate that ZGAN surpasses leading methods in generative modeling.

**Strengths:**

The article introduces a novel loss function, called zephyr loss, for GANs, which is referred to as ZGAN. This loss function is convex, smooth, and Lipschitz continuous, designed to enhance robustness and provide flexible gradient control.

**Weaknesses:**

1. The empirical results do not strongly support the claims made in the article. For instance, in Figure 4, the visual results of your ZGAN are unsatisfactory and significantly worse than those of state-of-the-art GAN models.

2. The article makes a mistake: even if the loss function is convex, smooth, and Lipschitz continuous, this does not ensure that the training process of GAN is Lipschitz continuous and dynamically stable. I recommend that the author include more analysis on how zephyr loss can stabilize GAN training and ensure that ZGAN meets Lipschitz continuity.

3. The article does not provide proof of dynamic stability near Nash equilibrium points. I advise the author to add this proof about local stability analysis around equilibrium points.

**Questions:**

1. What is the reason this loss function is called zephyr loss? Is there any prior research on this function? I recommend that the author provide more clarifications regarding related work on this function.

2. The empirical results lack metric analyses of sample diversity, such as FID, KID, or LIPIPS.  I suggested that the authors include these specific metrics in their evaluation, as these are very important results for generative models.

3. The empirical results do not include an analysis of ZGAN on high-resolution datasets. I suggested that the authors include the LSUN, CeleBA, or  ImageNet datasets in high resolution to demonstrate the effectiveness of ZGAN.

4. The empirical results do not explain how to convert the diffusion GAN into the diffusion ZGAN. I recommend the author add some description about this in the article.

5. The empirical results do not provide an ablation study about the parameters $\epsilon$ and $\alpha$.  I recommend the author add some ablation studies about these two important parameters.

6. In Figure 2, the gradient of zephyr loss exhibits a mutation near the center, whereas the least square loss remains smooth overall. This indicates the unstable gradient of zephyr loss. I suggest the author discuss the implications of this gradient behavior on training stability and performance.

---

### Official Review · Reviewer_4dgu · 2024-11-04

**Soundness:** 2
**Presentation:** 3
**Contribution:** 1
**Rating:** 3
**Confidence:** 3

**Summary:**

This work proposes using a smooth approximation to the Huber loss function, termed the 'zephyr loss', as part of GAN training. Authors show in theorem 4 that when the discriminator is optimal at each training step, minimizing the GAN objective is equivalent to minimizing a distance function that is close to the total variation distance. Experiments are provided demonstrating improved inception scores on cifar10, cifar100, STL10, and svhn datasets when compared to WGAN and LSGAN

**Strengths:**

* Presentation of the loss function is clear, and paper is well organized

**Weaknesses:**

* Much of the theory presented in this work seems incremental (especially Theorems 1-3), essentially reproducing the original convergence results of Goodfellow et al 2016 and Nowozin et al 2016 for a specific loss function. Accordingly, there is the same limitation that convergence results assume the discriminator is fully trained at each step, which is not realistic in practice.

* Concerningly, I do not see any references to the Huber loss or f-gan related papers [1,2]. It seems to me that this method is most likely a specific instance of an f-gan, and would benefit from analysis under this framework.

* While the experimental results seem promising, these datasets are fairly simple and I am not convinced that hyperparameter tuning will not be an issue for more complex datasets.


[1] Nowozin, Sebastian, Botond Cseke, and Ryota Tomioka. "f-gan: Training generative neural samplers using variational divergence minimization." Advances in neural information processing systems 29 (2016).

[2] Kurri, Gowtham R., et al. "α-GAN: Convergence and estimation guarantees." 2022 IEEE International Symposium on Information Theory (ISIT). IEEE, 2022.

**Questions:**

* What are the number of steps each method is trained for in Table 1?

---

### Official Review · Reviewer_Nk4F · 2024-11-04

**Soundness:** 1
**Presentation:** 2
**Contribution:** 1
**Rating:** 1
**Confidence:** 4

**Summary:**

Motivated by increasing the stability of GAN training, the paper proposes an alternative objective that is close to that of Least-Squares GAN (LSGAN) but where the squared loss is replaced by a soft-L1 loss that resembles the Huber loss. The generator’s objective is also reportedly replaced by a form of expected feature matching (as in Salimans et al. 2016a). Experiments on small image datasets show improvements in inception scores.

**Strengths:**

Originality and significance: To the best of my knowledge this is the first work to propose this type of GAN training objective. Improving stability of GAN training by increasing robustness properties via losses more robust to outliers and with bounded gradients is an interesting idea worth investigating.

Clarity and Quality: The paper is well written, and the approach is properly related and contrasted with the related GAN literature. Motivations are clearly stated. The algorithm is well described and easy to understand. The paper provides both a theoretical analysis of the algorithm’s property and quantitative and qualitative empirical evaluation.

**Weaknesses:**

The main weakness of the paper is what I believe may be an unsound equation derivation (in Eq 2 and 4) which invalidates the proof that the stated generator’s optimization objective approximately minimizes total loss variation, and thus invalidates the larger part of the theoretical analysis.
Moreover I think that how the generator’s objective is implemented in the code does not even correspond to what is stated in the paper (see question section for details).

The Zephyr-loss looks very close to a Huber loss, this connection is neither referenced nor discussed.

Another weakness is that the paper claims benefits of the approach – compared to its closest parent LSGAN – but without providing evidence that LSGAN training actually suffers from the ills that ZephyrGAN purports to cure, s.a. Gradient (in)stability and (insufficient) robustness to outliers. It would have been more convincing to measure and highlight these failures/difficulties, to show that they are indeed represent in LSGAN and then cured by ZephyrGAN.
One claimed advantage, “Flexibility in tuning with alpha”, clearly isn’t one, since alpha plays the same role as the learning rate (which is present in all other GAN variants).

Lastly, the difference in generation quality cannot really be assessed qualitatively on low resolution images (CIFAR-10, STL-10, SVHN). These experiments are much too low scale and quality for claiming the superiority of a generative modeling approach by nowadays’ standards.

**Questions:**

Q1: Can you relate and contrast your zephyr loss to the classical Huber loss used in robust regression? This seems very relevant and should be added to the related works discussion.
The LSGAN squared loss would also be smooth and convex, right?

Q2: It seems to me that the last two rows of Eq. 2 are not equivalent (absolute value of integral is not the same as integral of absolute value). And your code in supplementary seems to be doing something other than either of these two.

Taking epsilon=0 for simplicity, here is what I see:
* first row of eq.2: abs difference of expectations of discriminator-values

$ | E_{x \sim p_r}[D(x)] - E_{x \sim p_g}[D(x)] | $

$= | \int p_r(x) D(x) - \int p_g(x)D(x) .dx | $

$= | \int D(x) (p_r(x)-p_g(x)) .dx | $

* Last row of eq2: expected abs difference of weighted pdf (weighted by discriminator-values)  i.e. weighted total variation distance.

$\int D(x) |p_r(x) - p_g(x)| . dx$

* Implementation (g_loss line 106) : expected abs difference between discriminator-values at true point and generated point:

$E_{x \sim p_r, x’ \sim p_g} [ |D(x)-D(x’)| ] = \int \int p_r(x) p_g(x’) | D(x) - D(x’) |  .dx .dx’ $

Why should these 3 be the same?

Q3: The same question appears in the proof of Theorem 4 in Eq 4. when going from first to second line. If this doesn’t work, then the proof of the claim that the criterion leads to approximately minimizing weighted total variation breaks. But I don’t think this is what the code is doing anyway, right?

Q4: Instead of the above, did you try / compare with simply using the same kind of objective as LSGAN for the generator also, just replacing it by your zephyr loss?



Minor suggestions for improving clarity:

In 3.2. (around l 155) Say when you introduce them what values you give to t and f.
Also state what the output dimension of the discriminator D is.

L 210. “For values of a in equation 1 within the range $-\sqrt{\epsilon} < a > \sqrt{\epsilon}$, the loss behaves more like the absolute difference” ->
  “For values of a in equation 1 *OUTSIDE* the range $[-\sqrt{\epsilon}, \sqrt{\epsilon}]$, the loss behaves more like the absolute difference”

L 263 “For $p_r, p_g \in (0, 1]$”  there is no reason why these probability densities should always be <= 1.

---

### Meta-Review · Area_Chair_78hU · 2024-12-25

**Metareview:**

The paper propose a form of soft $l_1$ loss for training GANs implying robustness to outliers. The paper is not ready for publishing and all reviewers agreed on that.

Specifically:
* A big error in the derivation was pointed out in the review and acknowledged by the authors.
* A discrepancy between the method in the paper and the code provided
* weak experimentation and comparison to diffusion GAN, WGAN and LSGAN and moden implementations of GANs

We encourage the authors to take into account reviewers suggestions.

**Additional Comments On Reviewer Discussion:**

All reviewers discussed the papers with the authors and concluded that the paper is not in shape for publication,

---

### Decision · Program_Chairs · 2025-01-22

Reject